# TKG-LLM: Temporal Knowledge Graph as Enhanced Prompt Learning with LLM for Time Series Forecasting

## Abstract

Large Language Models (LLMs) based on Transformer have shown advantages in various domains by their powerful representation learning and context understanding capabilities. Recently, researchers have begun to explore their applications in time series forecasting. Although existing methods can achieve cross-modality embedding of time series into LLMs, the self-attention in LLMs is essentially "token-to-token". Position encoding can only reflect the sequential relationships between tokens, and the model cannot capture the temporal dependencies and correlations between features, thus not achieving excellent forecasting accuracy. Therefore, we propose the Temporal Knowledge Graph with LLM (TKG-LLM), which innovatively designs the TKG to capture the temporal structural information. We first build the TKG containing temporal edges to capture dependencies between time series and feature edges to capture relationships between features. Next, we apply the Graph Convolution Network (GCN) to encode the graph, generating node embeddings rich in temporal structural information. Finally, we fuse the time series embeddings with the graph node embeddings to enhance representational capabilities and utilize the enhanced embeddings for dynamic prompt selection to improve forecasting performance. Additionally, to better capture the multi-scale characteristics of time series and thereby improve the accuracy of forecasts. The time series is decomposed into three components: trend, season, and residual through Wavelet Decomposition (Daubechies 4) into TKG-LLM to capture multi-scale temporal features and sudden changes accurately. We demonstrate through visualizing experimental results that Wavelet Decomposition exhibits superior performance when dealing with non-stationary time series. Our empirical experiments on multiple benchmark datasets demonstrate that the proposed TKG-LLM achieves superior forecasting performance compared to baselines. Furthermore, our ablation experiment results verify the effectiveness of using the Temporal Knowledge Graph as enhanced prompt learning.

## 1 INTRODUCTION

Recent rapid advances in Large Language Models (LLMs) are driven by self-supervised learning based on the Transformer architecture (Vaswani et al., 2017). Training on massive text datasets with large-scale computational power results in models that possess highly general language representation capabilities. For example, the GPT (Radford et al., 2018) and LLaMA (Touvron et al., 2023) families exhibit not only significant advantages in traditional Natural Language Processing (NLP) tasks such as text classification (Devlin et al., 2019) and machine translation (Hendy et al., 2023), but also exhibit strong performance across various specialized domains, including code generation (Yadav & Mondal, 2025), medical informatics (Singhal et al., 2023), and financial analysis (Wang et al., 2023). With the successful utilization of LLMs across multiple domains, researchers have begun exploring how to migrate these models—leveraging their powerful generalization capabilities and contextual understanding—to time series analysis tasks. One Fits ALL (OFA) (Zhou et al., 2023) propose the Frozen Pretrained Transformer (FPT) framework, which fully freezes the self-attention layer and feedforward network layer of pre-trained models, requiring only fine-tuning of the embedding layer, layer normalization, and output linear layer. This allows time series models to

directly reuse large-scale pre-training achievements from NLP or CV without the need to separately train foundational models. Time-LLM (Jin et al., 2023) proposes Reprogramming to transform time series forecasting into language tasks that LLMs can handle. To further enhance the LLM's reasoning capabilities for time series, Prompt-as-Prefix (PaP) is designed to incorporate natural language prompts as input prefixes containing key information such as dataset background and task commands. Similarly, TEMPO (Cao et al., 2024) and PromptCast (Xue & Salim, 2024) are designed to guide LLMs in adapting to time series tasks through prompt learning. However, existing time series analysis methods based on LLMs mostly employ template-based fixed prompts. This prompting approach fails to precisely capture the dynamic characteristics of time series, making it difficult for the model to understand complex temporal structural information and resulting in insufficient prediction accuracy. Therefore, we innovatively propose the Temporal Knowledge Graph with LLM (TKG-LLM), which uses the TKG (Cai et al., 2022) as enhanced prompt learning. Specifically, we first design TKG to build the graph structure for time series, effectively capturing temporal dependencies and feature correlations. Next, we apply the Graph Convolutional Network (GCN) to encode this graph structure, generating node embeddings. Subsequently, we fuse these graph node embeddings with time series embeddings using multi-head attentions to generate enhanced embeddings. This design enables TKG-LLM to fully capture temporal features, thereby enhancing the model's forecasting performance. Comparisons with other baselines across several benchmark datasets show that TKG-LLM displays superior forecasting performance.

Before undertaking time series forecasting tasks, we must first complete the crucial step of time series decomposition. This is because, unlike other sequential data such as text, time series often exhibit complex temporal patterns (Nazerfard et al., 2010) comprising three key components: trend, seasonality, and residual. If the original time series is fed directly into the model, these components overlap, making it difficult for the model to learn the dynamics of each component independently. Furthermore, as pointed out in TEMPO (Cao et al., 2024), the self-attention layers of pre-trained models like GPT cannot learn Principal Component Analysis (PCA). When fed directly with the original time series, they cannot decouple the non-orthogonal trend and seasonal signals, leading to the model missing key patterns and reduced forecasting accuracy. TEMPO (Cao et al., 2024) first proposes using Seasonal-Trend Decomposition using Loess (STL) to decompose time series into additive components to capture the distinctive temporal patterns within each component. Unlike the above method, our TKG-LLM considers applying Wavelet Decomposition (Chernick & Michael, 2001) to decompose time series into three components—trend, seasonal, and residual—to better handle non-stationary time series. Through this decomposition method, TKG-LLM can better understand and forecast different features within the time series. Through comparative experiments and visualizations, we verify that Wavelet Decomposition can have more advantages than the STL decomposition (Cleveland & Cleveland, 1990) used in Tempo when dealing with time series such as $ETTh_1$, which exhibits more sudden fluctuations. Overall, our contributions are as follows:

(1)We apply Wavelet Decomposition for multi-resolution analysis, feature extraction, noise removal, and trend and periodicity analysis, enabling TKG-LLM to more accurately capture the temporal patterns of time series and further enhance the model's forecasting performance.

(2)We built the Temporal Knowledge Graph to enhance prompt learning. The enhanced prompt incorporates structural features of time series, enabling the prompt learning module to more accurately select prompts relevant to the input embeddings, thereby improving the model's forecasting ability.

(3)We make experiments and analysis on several benchmark datasets, and the results show that TKG-LLM displays superior short-term, long-term, and few-shot forecasting performance compared to other baselines.

(4)We prove through visualization experiments that Wavelet Decomposition has a better performance than STL decomposition when dealing with the non-stationary time series. Furthermore, ablation studies demonstrate the effectiveness of using the Temporal Knowledge Graph to enhance prompt learning.

## 2 RELATED WORK

Time series forecasting, as a critical task in time series data analysis, holds the potential to be widely utilized in the meteorology, energy, and transportation domains. Traditional time series forecasting

typically relies on statistical methods such as ARIMA (Anwar et al., 2016) and Exponential smoothing (Taylor & W, 2003) to capture autocorrelation and trends within the time series for forecasting. However, time series in real-world scenarios exhibit non-stationarity, high noise levels, and complex periodicity, making the limitations of traditional statistical methods in terms of model assumptions and flexibility increasingly apparent. The rise of machine learning and deep learning has provided new methods for time series forecasting. Recurrent Neural Networks (Dera et al., 2024)(RNN) and Long Short Term Memory (Myint & Khaing, 2025)(LSTM) can automatically learn long-term dependencies in time series. Nevertheless, these methods still face challenges such as high training data requirements and poor explainability. Recently, leveraging the powerful sequence modeling and knowledge migration capabilities of LLMs to solve time series forecasting problems has become a research hotspot in cross-mode learning and time series. LLMs demonstrate significant advantages in time series research due to their exceptional learning and representation capabilities. In time series forecasting tasks, LLMs have displayed significant advantages in forecasting accuracy compared to traditional statistical and machine learning methods (Sun et al., 2024a; Zhou et al., 2023; Cao et al., 2024). These studies primarily focus on achieving preliminary adaptations between time series and the semantic space of LLMs through input embeddings, while simultaneously exploring prompt learning to enhance the reasoning capabilities of LLMs regarding time series. For example, OFA (Zhou et al., 2023) divides the time series into patches and converts them into tokens, which are then directly input into a partially frozen GPT-2 model for fine-tuning. This fine-tuning targets only positional embeddings, layer normalization, input embeddings, and the output linear layer. Meanwhile, it mitigates data distribution shifts through instance normalization and extracts local semantics via patching operations, thereby achieving adaptation between the time series and the pre-trained model. TEST (Sun et al., 2024a) also patches time series using sliding windows, then generates temporal embeddings aligned with the embedding layer space of LLM. Finally, it trains soft prompts to guide frozen LLM in understanding these time series embeddings, thereby accomplishing time series tasks. TEMPO (Cao et al., 2024) attempts to combine temporal decomposition with semi-soft prompting design. It proposes using STL decomposition to separate trend, seasonal, and residual components, thereby enhancing the model's effectiveness in dealing with time series. Different prompts are generated for each of these three components to improve the model's adaptability when processing time series. The above methods have promoted the development of LLMs in the domain of time series analysis, laying a crucial foundation for our subsequent research. However, these methods merely consider cross-modal injection of time series into LLM for inference, failing to fully explore the complex temporal dependencies and patterns hidden within time series, which leads to imprecise forecasting. Therefore, we propose TKG-LLM, a prompt learning method enhanced by Temporal Knowledge Graph to fully capture the temporal structure information of time series. By fusing the structural features of time series, this method enables the prompt learning module to select prompts more accurately based on the series characteristics, ultimately significantly improving forecast performance.

Time series decomposition is a method that makes the long-term trend, seasonal variations, and random fluctuations within an original series more clearly identifiable and modeled by a forecasting model, thereby significantly improving overall forecasting performance. In practical applications, time series often exhibit non-stationary characteristics and contain complex dynamic changes. Therefore, how to effectively decompose these components has become a crucial research focus (Huang, 2000). Tempo (Cao et al., 2024) proposes using STL decomposition (Rojo et al., 2017) to divide the time series into three components: the trend component, the seasonal component, and the residual component. STL is based on Localized Ordinary Least Squares regression (Cleveland & Devlin, 1988) to achieve smooth decomposition, showing strong robustness and the ability to deal with various time-series patterns. However, STL generally assumes smooth characteristics for trend and seasonal components, showing limitations when dealing with high-frequency sudden changes, local anomalies, or complex periodic structures. To overcome these limitations, a series of improved methods have been proposed, including Wavelet Decomposition (Wang & Wang, 2010). This method originates from the domain of signal processing and is particularly well-suited for industrial condition monitoring and fault diagnosis scenarios. Its core principle involves decomposing the original signal into various frequency subband components through multiscale analysis, thereby effectively capturing transient features, local sudden changes, and multi-resolution details within non-stationary series. Compared to STL, Wavelet Decomposition does not require the pre-assumption of global periodicity. It can adaptively extract local time-frequency features by flexibly selecting wavelet basis functions and decomposition levels. Consequently, it shows superior perfor-

mance in time series tasks featuring complex periodic structures or sudden changes. We consider applying Wavelet Decomposition for time series decomposition in TKG-LLM. Detailed operations are described in Section 3.2. The visualization results in Section 4.4 demonstrate that Wavelet Decomposition outperforms STL decomposition when dealing with sudden change data such as the $ETTh_1$ dataset.

# 3 METHODOLOGY

Figure 1 shows the architecture of TKG-LLM. Its contains four crucial modules: 1) A time-series decomposition module based on Wavelet Decomposition, decomposing trend, seasonal, and residual components; 2) A Temporal Knowledge Graph building, capturing temporal structure information to enhance prompt learning; 3) A dynamic prompt learning module, selecting the most relevant prompts; 4) A backbone model based on GPT-2 as a pre-trained Large Language Model (LLM), generating time series forecasts.

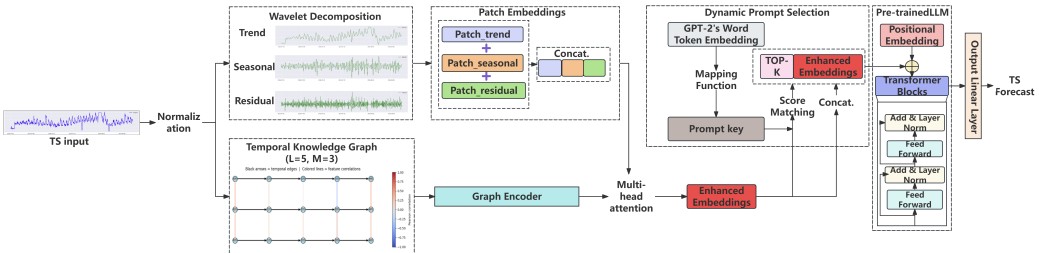

Figure 1: TKG-LLM Architecture.

## 3.1 PROBLEM STATEMENT

Given a multivariate time series $X \in \mathbb{R}^{N \times T}$, where $N$ represents the number of variables, $T$ represents the time steps, $X_{:,t} \in \mathbb{R}^{N \times 1}$ represents the time series for all variables at time step $t$, and $X_{i,:} \in \mathbb{R}^{1 \times T}$ represents the time series for the $i$ variable. This paper aims to construct a forecasting module $F(*)$ based on a historical observation window $t_{forecast}$ to forecast changes in series values for the next $t'_{forecast}$ time steps. Specifically, for an initial time step $t$, the forecasting task can be formulated as:

$$Y' = X'_{:,t:t+t'_{forecast}-1} = F(X_{:,t-t_{forecast}:t-1}) \tag{1}$$

where $Y' \in \mathbb{R}^{N \times \tau'}$ is the forecasted output, $X_{:,t-\tau:t-1} \in \mathbb{R}^{N \times \tau}$ is the input history window.

## 3.2 WAVELET DECOMPOSITION

To better capture features across different time scales and thereby enhance overall forecasting performance, we employ Wavelet Decomposition to decompose the time series. Specifically, the Daubechies 4 wavelet is utilized. The formula for calculating its wavelet coefficients is as follows:

$$c_{j,k} = \sum_{n=-\infty}^{\infty} x[n] \cdot \psi_{j,k}[n] \tag{2}$$

where $c_{j,k}$ is the wavelet coefficient, in which $j$ is the decomposition level and $k$ is the time index, and $\psi_{j,k}[n]$ is the discrete wavelet basis function.

First, we apply Wavelet Decomposition to the original time series, decomposing it into three components: trend, seasonal, and residual. Each time series undergoes the decomposition, which results in a set of coefficients. This set includes an approximation coefficient $cA$ representing the low-frequency part (trend) of the series, and multiple detail coefficients $cD$ representing the high-frequency portion (seasonal and residual). Next, we reconstruct the three components: the trend

component is reconstructed using the approximation coefficient, the seasonal component is reconstructed using the first-order detail coefficient, and the residual component is obtained by subtracting the trend and seasonal components from the original time series. Finally, we contact the three components together:

$$T(t) = \sum_k cA_{j,k}\phi_{j,k}(t) \tag{3}$$

$$S(t) = \sum_j \sum_k cD_{j,k}\psi_{j,k}(t) \tag{4}$$

$$R(t) = X(t) - T(t) - S(t) \tag{5}$$

where $T(t)$, $S(t)$, and $R(t)$ are the trend, seasonal, and residual components, respectively. $\phi_{j,k}$ is the scale function, $cD_{j,k}$ is the detail coefficient, and $\psi_{j,k}(t)$ is the wavelet basis function. In our experiments, we will compare the Wavelet Decomposition used in TKG-LLM with the STL decomposition used in TEMPO to prove the advantages of Wavelet Decomposition when dealing with non-stationary time series. After completing the decomposition, we need to perform patch splitting on each component using a sliding window. Finally, these patches are concatenated together.

## 3.3 TEMPORAL KNOWLEDGE GRAPH-ENHANCED PROMPT LEARNING

Time series inherently have structural information such as temporal dependencies and correlations between features. Temporal knowledge graphs can explicitly capture this structural information, helping models better understand relationships within time series. We first need to build the Temporal Knowledge Graph, which transforms time series into a graph structure. This graph enables the representation of temporal dependencies (temporal edges) and feature correlations (feature edges). The constructed graph primarily consists of nodes and edges, where each feature value at every time point represents a node. Assuming the time series $X \in \mathbb{R}^{N \times T}$, where $T$ is the series length and $N$ is the number of features, resulting in a total of $N \times T$ time nodes $V$. The edges contain both temporal edges and feature edges. Temporal edges are directed connections established between the same feature at adjacent time steps. The set of edges is $E_{\text{temporal}} = \{((t \cdot N + f, (t + 1) \cdot N + f) \mid 0 < t < T - 1, 0 < f < N\}$, with a weight of 1 indicating deterministic temporal dependency. Feature edges are undirected connections established between different features at the same time step. The set of edges is $E_{\text{feature}} = \{(t \cdot N + f_i, t \cdot N + f_j) \mid 0 < t < T, 0 < f_j < f_i < N\}$, with weights corresponding to Pearson correlation coefficients, the formula as:

$$\rho_{f_i,f_j} = \frac{\text{cov}(X_{f_i}, X_{f_j})}{\sigma_{f_i}\sigma_{f_j}} \tag{6}$$

For each time step $t$ we estimate the local Pearson correlation between features $f_i$ and $f_j$. Subsequently, we apply the Graph Convolutional Network (GCN) to encode the Temporal Knowledge Graph, generating node embeddings. The specific formula is as follows:

$$H^{(l+1)} = \sigma\left(\tilde{D}^{-\frac{1}{2}}\tilde{A}\tilde{D}^{-\frac{1}{2}}H^{(l)}W^{(l)}\right) \tag{7}$$

where $H^{(l+1)}$ is the node feature representation of layer $l + 1$, where $\sigma$ is the ReLU activation function. $\tilde{D}^{-\frac{1}{2}}\tilde{A}\tilde{D}^{-\frac{1}{2}}$ is the normalized symmetric adjacency matrix(Among these, $\tilde{A}$ is the weighted adjacency matrix, $\tilde{A} \in \mathbb{R}^{V \times V}$; $D$ is the degree matrix, $\tilde{D}_{ii} = \sum_j \tilde{A}_{ij}$), $H^{(l)}$ is the node feature vector for layer $l$, and $W^{(l)}$ is the weight matrix that can be trained for layer $l$. Following this, we fuse the node embeddings with the time series embeddings using the Multi-Head Attention to generate enhanced embeddings. The following formula is provided for multi-head attention fusion:

$$Q = E_{\text{ts}}W_Q \tag{8}$$

$$K = E_{\text{graph}}\mathbf{W}_K \tag{9}$$

$$V = E_{\text{graph}}\mathbf{W}_V \tag{10}$$

$$head_h = softmax\left(\frac{Q_hK_h^T}{\sqrt{d_k}}\right)V_h \tag{11}$$

where $W_Q, W_K, W_V \in \mathbb{R}^{d \times d}$, $d$ is the embedding dimension, $d_k = d/H$, $h$ is the number of heads.

## 3.4 DYNAMIC PROMPT LEARNING

The key to this prompt selection module is to dynamically choose the most relevant prompts from the prompt pool based on the semantic features of the embeddings. These prompts are guiding the large language model to generate more accurate forecasts. TKG-LLM generates a global semantic representation of the embeddings for subsequent similarity comparisons by computing operations such as the average, maximum, or weighted sum of the maximum and average values of the embeddings. Secondly, we need to normalize the prompt key (the prompt key is obtained through the word embedding matrix of GPT-2 and linear layer mapping) and enhanced embedding, calculate the similarity matrix between the normalized enhanced embedding and the prompt key, and select the $k$ prompts with the highest similarity according to the similarity matrix. Finally, we will splice the selected $k$ prompts with enhanced embedding to form the final embeddings.

## 3.5 TIME SERIES FORECASTING BASED ON GPT-2

The decoder-only architecture of GPT-2 (Radford et al., 2019) is better suited for the sequence generation requirements of time series compared to other LLMs. TEST (Sun et al., 2024a) explicitly points out that the generative structure of GPT-2 as the backbone model achieves outstanding performance in temporal forecasting tasks. For the time series forecasting task in this paper, to reduce training complexity while preserving the pre-trained model's general feature extraction capabilities, only a series of parameters (layer normalization and positional encoding) was trained (Lu et al., 2022), with the remainder frozen. Research findings indicate that fine-tuning methods—where most parameters remain frozen—typically yield superior performance compared to fully retraining large language models (Houlsby et al., 2019). We apply the above method to obtain the final embeddings, which are then input into the GPT-2 model to generate forecasts.

# 4 EXPERIMENTS

This paper compares our proposed TKG-LLM with several baselines on five benchmark datasets. Through a series of experiments, we prove the effectiveness of TKG-LLM in various time series forecasting tasks: short-term forecasting (Section 4.1), long-term forecasting (Section 4.2), and few-shot forecasting (Section 4.3). Sections 4.4 and 4.5, respectively, propose comparative studies on temporal decomposition and ablation studies. Throughout the experiments, all models adhere to the established experimental configurations for the respective research tasks (Liu et al., 2024).

**Dataset.** We run short-term forecasting experiments using the M4 dataset (Makridakis et al., 2020), and run long-term forecasting experiments on four datasets—$ETTh_1$, $ETTh_2$, $ETTm_1$, and $ETTm_2$ from the widely used Electric Power Transformer Temperature (ETT) dataset (Zhou et al., 2021).

**Baselines.** We select LLM-based methods including OFA (Zhou et al., 2023), Tempo (Cao et al., 2024), and TEST (Sun et al., 2024b), and a group of Transformer-based methods , including PatchTST (Nie et al., 2022), iTransformer (Liu et al., 2024), and Autoformer (Wu et al., 2021). Additionally, we select DLinear (Zeng et al., 2023) and TimesNet (Wu et al., 2023). An introduction to the above eight baselines is found in Appendix A.1.

**Evaluation Metrics.** Our experiments apply Symmetric Mean Absolute Percentage Error (SMAPE), Mean Absolute Scaled Error (MASE), and Overall Weighted Average (OWA) to evaluate the performance of short-term forecasts. During experiments on long-term forecasting using the Mean Squared Error (MSE) and Mean Absolute Error (MAE). The specific calculation formulas for each evaluation metric are provided in Appendix A.2.

## 4.1 SHORT-TERM FORECASTING

**Setup.** Our experiment evaluates the performance of TKG-LLM on short-term forecasting tasks using the M4 dataset. The forecast horizon was set to [6,48], with the input time series length doubled to match the forecast horizons. For evaluation metrics, the Mean Absolute Percentage Error (SMAPE), Mean Absolute Scaled Error (MASE), and Overall Weighted Average (OWA) were selected.

Table 1: TKG-LLM and other baselines on the M4 dataset, with short-term forecasting metrics results under the forecast horizon [6,48]

| Method | TKG-LLM | OFA | iTransformer | Dlinear | PatchTST | TimesNet | Autoformer |
|---|---|---|---|---|---|---|---|
| **Yearly** | | | | | | | |
| SMAPE | 13.596 | 15.110 | 13.652 | 16.965 | 13.477 | 15.378 | 13.974 |
| MASE | 3.068 | 3.565 | 3.095 | 4.283 | 3.095 | 3.554 | 3.134 |
| OWA | 0.802 | 0.911 | 0.807 | 1.058 | 0.807 | 0.918 | 0.822 |
| **Quarterly** | | | | | | | |
| SMAPE | 10.529 | 10.597 | 10.353 | 12.145 | 10.380 | 10.465 | 11.338 |
| MASE | 1.236 | 1.253 | 1.209 | 1.520 | 1.233 | 1.227 | 1.365 |
| OWA | 0.929 | 0.938 | 0.911 | 1.106 | 0.921 | 0.923 | 1.012 |
| **Monthly** | | | | | | | |
| SMAPE | 12.870 | 13.258 | 13.079 | 13.514 | 12.959 | 13.513 | 13.958 |
| MASE | 0.958 | 1.003 | 0.974 | 1.037 | 0.970 | 1.039 | 1.103 |
| OWA | 0.896 | 0.931 | 0.911 | 0.956 | 0.905 | 0.957 | 1.002 |
| **Others** | | | | | | | |
| SMAPE | 5.594 | 6.124 | 4.780 | 6.709 | 4.952 | 6.913 | 5.485 |
| MASE | 3.626 | 4.116 | 3.231 | 4.953 | 3.347 | 4.507 | 3.865 |
| OWA | 1.160 | 1.259 | 1.012 | 1.487 | 1.049 | 1.438 | 1.187 |
| **Avg** | | | | | | | |
| SMAPE | 12.111 | 12.690 | 12.142 | 13.693 | 12.059 | 12.880 | 12.909 |
| MASE | 1.643 | 1.808 | 1.631 | 2.095 | 1.623 | 1.836 | 1.771 |
| OWA | 0.876 | 0.940 | 0.874 | 1.051 | 0.869 | 0.955 | 0.939 |

**Results.** Table 1 shows the evaluation metrics of TKG-LLM and several baselines on the M4 dataset across different forecast horizons, including the Mean Absolute Percentage Error (SMAPE), Mean Absolute Scaled Error (MASE), and Overall Weighted Average (OWA). In general, the short-term forecasting metrics of our proposed TKG-LLM are lower than all other baselines, so we can conclude that TKG-LLM displays more precise short-term forecasting performance. It is worth noting that both TKG-LLM and OFA are based on the GPT-2, which is fine-tuned on specific layers for time series forecasting. Table 1 shows that the TKG-LLM achieved an average MASE of 0.165 for short-term forecasting metrics, representing a 9.13% improvement in short-term forecasting accuracy compared to OFA. It also improves SMAPE and OWA by 4.56% and 6.81%, respectively.

## 4.2 LONG-TERM FORECASTING

**Setup.** Our experiment evaluates the performance of TKG-LLM on long-term forecasting using the ETT series sub-datasets (containing $ETTh_1$, $ETTh_2$, $ETTm_1$, and $ETTm_2$). The input time series length was set to 512, and the models' forecasting performance was evaluated for four forecasting horizons [96, 192, 336, 720]. For evaluation metrics, Mean Squared Error (MSE) and Mean Absolute Error (MAE) were selected as the metrics.

**Results.** Table 2 shows the Mean Squared Error (MSE) and Mean Absolute Error (MAE) metrics for several time series forecasting models on the ETT dataset across different forecast horizons. Lower metric values mean more accurate forecasts. Based on the average performance of the four ETT sub-datasets ($ETTh_1$, $ETTh_2$, $ETTm_1$, $ETTm_2$), TKG-LLM displays the lowest MSE and MAE values in most of the results, indicating its significantly superior forecasting accuracy compared to other baselines (such as iTransformer, DLinear, TimesNet, PatchTST, and Autoformer). Its forecasting performance is comparable to or even better than other LLM-based methods like OFA, Tempo, and TEST. As the forecasting steps increase from 96 to 720, TKG-LLM shows the smallest performance decrease, demonstrating its better forecasting capability for long series.

## 4.3 FEW-SHOT FORECASTING

**Setup.** Model performance was evaluated under the few-shot forecasting scenario, which aims to verify whether the model can still generate accurate forecasts when training data is limited. In this experiment, we selected the top 5% and top 10% of the data as training samples. For evaluation metrics, Mean Squared Error (MSE) and Mean Absolute Error (MAE) were selected.

**Results.** In few-shot forecasting (using 5% and 10% of training data), TKG-LLM shows strong generalization capabilities and stability on several ETT sub-datasets. As shown in Tables 3 and 4, all baselines display varying degrees of performance decrease as training data are reduced. However, TKG-LLM can maintain relatively low MSE and MAE values in most results, indicating that it still possesses strong time series forecasting capabilities under data-limited conditions. Particularly on the $ETTh_1$ and $ETTm_2$ datasets, TKG-LLM results in significantly lower mean errors than most baselines at 5% and 10% training data, further verifying its forecasting capability. Overall, TKG-LLM shows superior forecasting performance in few-shot forecasting tasks, making it suitable for scenarios where data availability is limited when being utilized in practical applications.

## 4.4 TIME SERIES DECOMPOSITION

To explore the decomposition performance on the $ETTh_1$ and $ETTm_1$ datasets, Figure 2 and 3 show a visual comparison between STL and Wavelet Decomposition. As shown in Figure 3, in the $ETTh_1$, the trend component from Wavelet Decomposition (green dashed line) exhibits greater sensitivity to local variations. In contrast, the trend component from STL decomposition (red solid line) appears smoother and is designed to overlook local features. The residuals from Wavelet Decomposition demonstrate smaller fluctuations and a more uniform distribution, indicating a more thorough decomposition. Conversely, the larger fluctuations in STL residuals suggest the potential presence of inadequately decomposed components. As shown in Figure 3, within the relatively stable $ETTm_1$ dataset, both methods exhibit distinct advantages: STL excels at extracting long-term stable trends and seasonal patterns, while Wavelet Decomposition performs better at preserving local features and handling residuals. In contrast, for the $ETTh_1$ dataset exhibiting non-stationary characteristics, Wavelet Decomposition demonstrates a clear advantage due to its superior handling of multi-scale features and local variations—a critical factor for subsequent forecasting tasks. Therefore, for forecasting non-stationary time series, Wavelet Decomposition can be prioritized.

## 4.5 ABLATION STUDIES

In ablation experiments, to verify the critical role of the Temporal Knowledge Graph (TKG) module in model performance, we compare the performance of the full TKG-LLM against the model with the TKG module removed on the $ETTh_1$ and $ETTm_1$ datasets (see Table 5). In ablation results, removing the TKG module increases MSE from 0.371 to 0.382 and MAE from 0.402 to 0.404 in the $ETTh_1$-96 forecasting task. For the $ETTm_1$-96 task, MSE rises from 0.289 to 0.292 and MAE from 0.345 to 0.349. Although the performance decrease appears numerically small, such errors become practically significant in time series forecasting tasks, particularly when utilized for high accuracy. This result shows that the Temporal Knowledge Graph effectively enhances the representation capability of prompt learning by explicitly constructing complex relationships between temporal points and features, improving the model's ability to capture temporal dependencies and feature correlations. Ablation experiments fully prove the necessity of the Temporal Knowledge Graph-enhanced prompt learning in the TKG-LLM, whose effective introduction promotes improved forecasting performance.

## 5 CONCLUSION

Our study innovatively proposes Temporal Knowledge Graph-enhanced prompt learning based on Large Language Model (LLM), while considering the decomposition of time series using Wavelet Decomposition. This method has achieved breakthrough progress in time series forecasting. First, Daubechies 4 decomposes non-stationary time series into trend, seasonal, and residual components, thereby accurately capturing multi-scale temporal features and local changes. This provides a high-quality foundation of temporal features for subsequent tasks. Second, TKG-LLM builds the Temporal Knowledge Graph to enhance prompt learning. This module transforms time series into graph

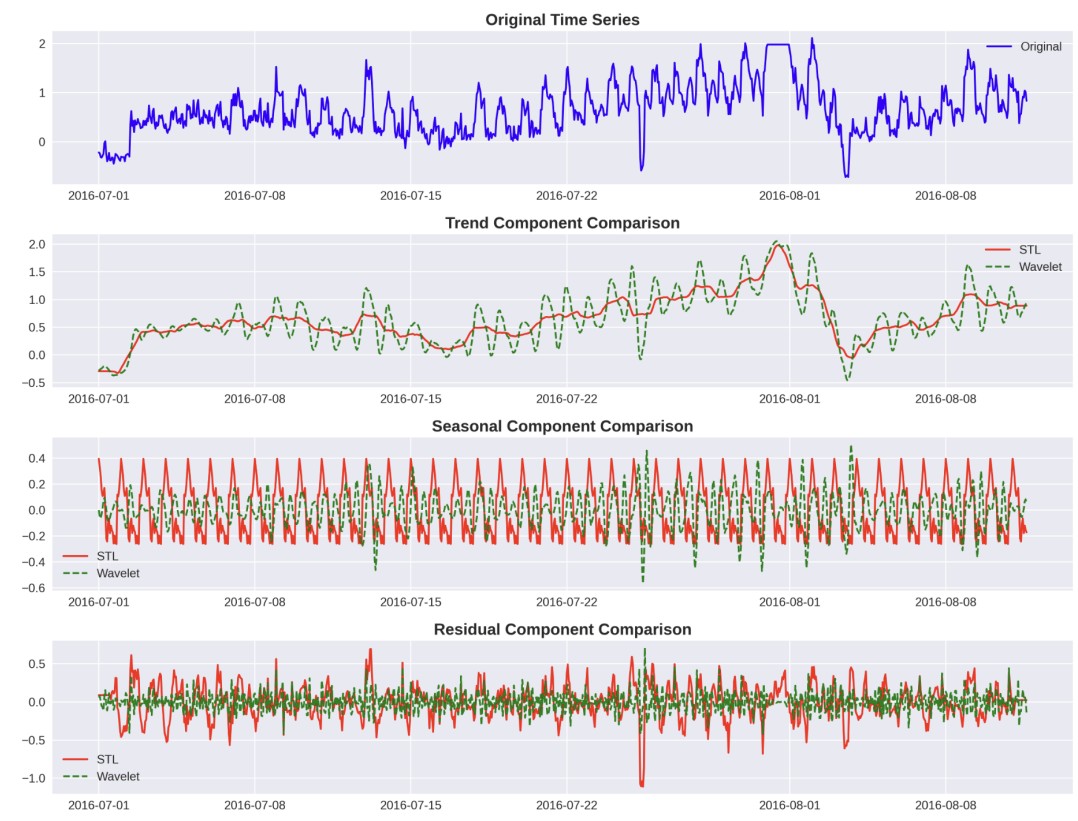

Figure 2: Comparison of Wavelet and STL Decomposition on the $ETTh_1$.

structures, where the graph reflects temporal dependencies (temporal edges) and correlations between features (feature edges). Subsequently, it will fuse these through the use of Graph Convolutional Networks and multi-head attentions to generate enhanced embeddings. Simultaneously, the dynamic prompt learning module selects the top prompts, thereby providing strong support for forecasting tasks. Concatenate the top-k prompt with the enhanced embeddings to generate the final embeddings, which are then fed into the model to generate forecasts. Finally, our experiments on short-term, long-term, and few-shot forecasting on several benchmark datasets fully validate the forecasting capabilities of TKG-LLM. In future practical utilization, TKG-LLM can deliver significant value across critical industries such as energy, finance, and healthcare. For example, in ICU vital sign monitoring within healthcare, the model decomposes trends (disease progression), seasonality (circadian rhythms), and residuals (sudden deterioration). By integrating implicit relationships among features like heart rate, blood pressure, and blood oxygen levels through the Temporal Knowledge Graph, it enables early warnings to shorten the golden intervention window.

## CODE AVAILABILITY

The code is available at: `https://anonymous.4open.science/r/TKG-LLM66-3A3B/`.

### AUTHOR CONTRIBUTIONS

If you'd like to, you may include a section for author contributions as is done in many journals. This is optional and at the discretion of the authors.

### ACKNOWLEDGMENTS

Use unnumbered third level headings for the acknowledgments. All acknowledgments, including those to funding agencies, go at the end of the paper.

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

## A EXPERIMENTAL DETAILS

We have provided code that can be reproduced and data in the supplementary materials for the system. To ensure a fair comparison with the selected baselines, we primarily followed the same experimental setup (Liu et al., 2024) and employed GPT2-small (Radford et al., 2019) as the backbone model. All experiments were implemented in PyTorch and conducted on a NVIDIA GeForce RTX 4070 GPU. We set the patch length to 16, the slide window to 8, and maintained the number of semantic anchor candidates to 1000.

### A.1 BASELINES

Below is a brief introduction to the eight baselines compared with TKG-LLM:

OFA (Zhou et al., 2023): This research proposes the Frozen Pretrained Transformer (FPT) framework, which leverages pre-trained NLP or CV models by freezing self-attention and feedforward layers while fine-tuning the adaptation layer. It achieves performance comparable to or surpassing SOTA across seven core tasks, including time series classification and forecasting.

Tempo (Cao et al., 2024): This research decomposes time series into three components—trend, seasonality, and residual—via STL decomposition, and designs a semi-soft prompting to adapt generative LLMs.

TEST (Sun et al., 2024b): This research explicitly categorizes the technical approaches for integrating Time Series (TS) with LLM into two types: LLM-for-TS and TS-for-LLM. It proposes using a three-layer contrastive learning mechanism to constrain the consistency between the TS embedding space and the LLM text embedding space.

Autoformer (Wu et al., 2021): This research proposes a deep decomposition architecture, embedding serial decomposition as an internal unit within the encoder-decoder structure of Autoformer. During forecasting, the model alternates between optimizing forecasts and performing serial decomposition.

PatchTST (Nie et al., 2022): This research has shown that Transformers are not only effective for time series forecasting but can also extract transferable representations through self-supervised learning by designing a "patch + channel" architecture.

DLinear (Zeng et al., 2023): This research decomposes raw input data into trend components using a moving average kernel and residual components. Two single-layer linear layers are then utilized on each component, and these features are concatenated to obtain the final forecast.

TimesNet (Wu et al., 2023): This research proposes a modular approach to modeling temporal variation. By transforming one-dimensional time series into a two-dimensional space, it can simultaneously represent intra-period and inter-period variations.

iTransformer (Liu et al., 2024): This research proposes treating the entire time series of a single variable as a single token and reversing the roles of self-attention and FNN mechanisms within the Transformer. Self-attention is employed to capture correlations between variables, while FNN is utilized for global representation within the sequence.

### A.2 EVALUATION METRICS

The calculations of evaluation metrics are as follows:

$$MSE = \frac{1}{H} \sum_{h=1}^{H} (Y_h - Y_h')^2 \tag{12}$$

$$MAE = \frac{1}{H} \sum_{h=1}^{H} |Y_h - Y_h'| \tag{13}$$

$$SMAPE = \frac{200}{H} \sum_{h=1}^{H} \frac{|Y_h - Y_h'|}{|Y_h| + |Y_h'|} \tag{14}$$

$$MASE = \frac{1}{H} \sum_{h=1}^{H} \frac{|Y_h - Y'_h|}{\frac{1}{H-s} \sum_{j=s+1}^{H} |Y_j - Y_{j-s}|} \tag{15}$$

$$OWA = \frac{1}{2} \left[ \frac{SMAPE}{SMAPE_{\text{Naive2}}} + \frac{MASE}{MASE_{\text{Naive2}}} \right] \tag{16}$$

### A.3 LONG-TERM FORECASTING RESULTS

The table 2 shows the long-term forecast results of TKG-LLM and other baselines on the ETT dataset.

### A.4 FEW-SHOT FORECASTING RESULTS

The table 3 and 4 below shows the few-shot forecast results for TKG-LLM and other baselines on the 5% and 10% ETT datasets.

### A.5 TIME SERIES DECOMPOSITION

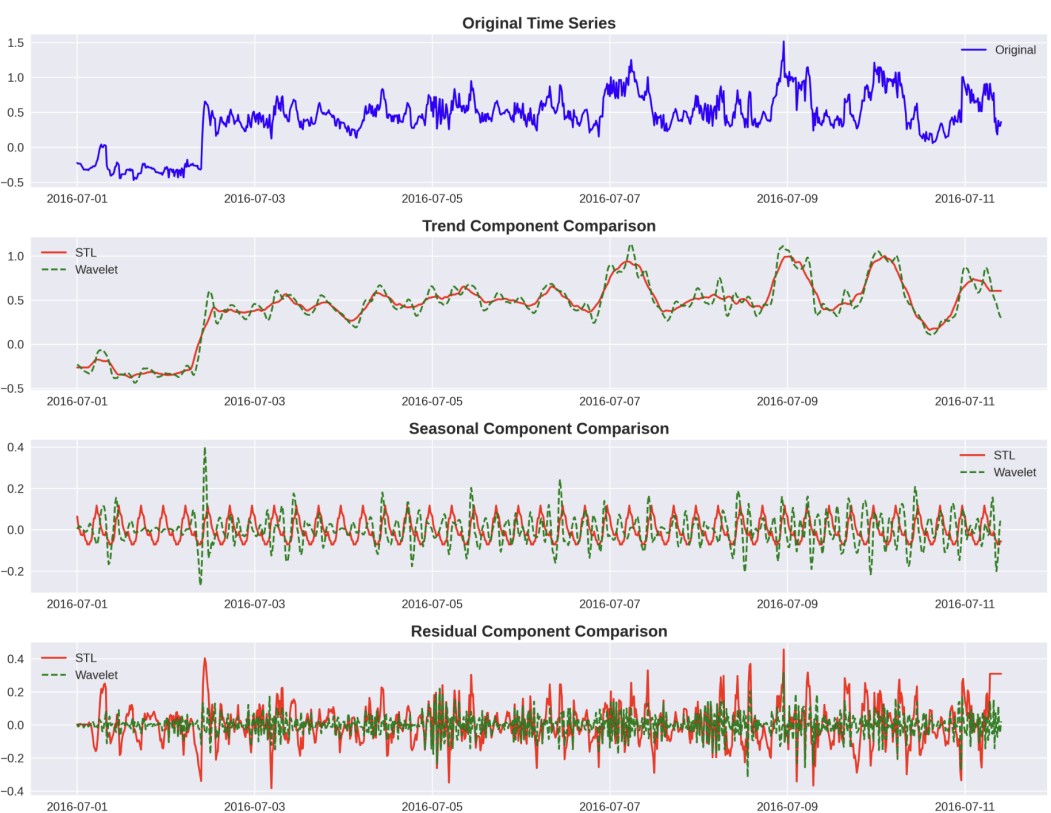

Figure 3: Comparison of Wavelet and STL Decomposition on the $ETTm_1$.

### A.6 ABLATION STUDIES

Table 2: Long-term forecast metrics results for TKG-LLM and other baselines on the ETT dataset, at forecast horizons of [96, 192, 336, 720]

| Dataset | Horizon | TKG-LLM | | OFA | | Tempo | | Test | | iTransformer | |
|---|---|---|---|---|---|---|---|---|---|---|---|
| | | MSE | MAE | MSE | MAE | MSE | MAE | MSE | MAE | MSE | MAE |
| $ETTh_1$ | 96 | 0.371 | 0.402 | 0.379 | 0.402 | 0.400 | 0.406 | 0.372 | 0.400 | 0.395 | 0.420 |
| | 192 | 0.425 | 0.440 | 0.415 | 0.424 | 0.426 | 0.421 | 0.414 | 0.422 | 0.427 | 0.441 |
| | 336 | 0.455 | 0.458 | 0.435 | 0.440 | 0.441 | 0.430 | 0.422 | 0.437 | 0.445 | 0.457 |
| | 720 | 0.529 | 0.504 | 0.441 | 0.459 | 0.443 | 0.451 | 0.447 | 0.467 | 0.537 | 0.530 |
| | Avg | 0.445 | 0.451 | 0.418 | 0.431 | 0.428 | 0.427 | 0.414 | 0.431 | 0.451 | 0.462 |
| $ETTh_2$ | 96 | 0.294 | 0.348 | 0.289 | 0.347 | 0.301 | 0.353 | 0.275 | 0.338 | 0.304 | 0.360 |
| | 192 | 0.375 | 0.397 | 0.358 | 0.392 | 0.355 | 0.389 | 0.340 | 0.379 | 0.377 | 0.403 |
| | 336 | 0.399 | 0.427 | 0.383 | 0.414 | 0.379 | 0.408 | 0.329 | 0.381 | 0.405 | 0.429 |
| | 720 | 0.418 | 0.444 | 0.438 | 0.456 | 0.409 | 0.440 | 0.381 | 0.423 | 0.443 | 0.464 |
| | Avg | 0.371 | 0.404 | 0.367 | 0.402 | 0.361 | 0.398 | 0.331 | 0.380 | 0.382 | 0.414 |
| $ETTm_1$ | 96 | 0.289 | 0.345 | 0.296 | 0.353 | 0.438 | 0.424 | 0.293 | 0.346 | 0.312 | 0.366 |
| | 192 | 0.335 | 0.362 | 0.335 | 0.373 | 0.461 | 0.432 | 0.332 | 0.369 | 0.347 | 0.385 |
| | 336 | 0.376 | 0.400 | 0.369 | 0.394 | 0.515 | 0.467 | 0.368 | 0.392 | 0.379 | 0.404 |
| | 720 | 0.420 | 0.427 | 0.418 | 0.424 | 0.591 | 0.509 | 0.418 | 0.420 | 0.441 | 0.442 |
| | Avg | 0.355 | 0.383 | 0.355 | 0.386 | 0.501 | 0.458 | 0.353 | 0.382 | 0.370 | 0.399 |
| $ETTm_2$ | 96 | 0.167 | 0.258 | 0.170 | 0.264 | 0.185 | 0.267 | - | - | 0.179 | 0.271 |
| | 192 | 0.228 | 0.296 | 0.231 | 0.306 | 0.243 | 0.304 | - | - | 0.242 | 0.313 |
| | 336 | 0.282 | 0.329 | 0.280 | 0.339 | 0.309 | 0.345 | - | - | 0.288 | 0.344 |
| | 720 | 0.364 | 0.385 | 0.373 | 0.402 | 0.386 | 0.395 | - | - | 0.378 | 0.397 |
| | Avg | 0.260 | 0.317 | 0.265 | 0.328 | 0.280 | 0.328 | - | - | 0.272 | 0.331 |

| Dataset | Horizon | DLinear | | TimesNet | | PatchTST | | Autoformer | |
|---|---|---|---|---|---|---|---|---|---|
| | | MSE | MAE | MSE | MAE | MSE | MAE | MSE | MAE |
| $ETTh_1$ | 96 | 0.414 | 0.421 | 0.468 | 0.475 | 0.516 | 0.485 | 0.613 | 0.552 |
| | 192 | 0.439 | 0.437 | 0.484 | 0.485 | 0.484 | 0.485 | 0.722 | 0.598 |
| | 336 | 0.463 | 0.464 | 0.536 | 0.516 | 0.536 | 0.516 | 0.750 | 0.619 |
| | 720 | 0.467 | 0.481 | 0.593 | 0.537 | 0.593 | 0.537 | 0.721 | 0.616 |
| | Avg | 0.445 | 0.451 | 0.520 | 0.505 | 0.520 | 0.505 | 0.702 | 0.596 |
| $ETTh_2$ | 96 | 0.334 | 0.389 | 0.376 | 0.415 | 0.376 | 0.415 | 0.413 | 0.451 |
| | 192 | 0.381 | 0.415 | 0.409 | 0.440 | 0.409 | 0.440 | 0.474 | 0.477 |
| | 336 | 0.471 | 0.482 | 0.425 | 0.455 | 0.425 | 0.455 | 0.547 | 0.543 |
| | 720 | 0.639 | 0.559 | 0.488 | 0.494 | 0.488 | 0.494 | 0.516 | 0.523 |
| | Avg | 0.456 | 0.461 | 0.425 | 0.451 | 0.425 | 0.451 | 0.488 | 0.499 |
| $ETTm_1$ | 96 | 0.624 | 0.522 | 0.329 | 0.377 | 0.329 | 0.377 | 0.774 | 0.614 |
| | 192 | 0.599 | 0.511 | 0.371 | 0.401 | 0.371 | 0.401 | 0.754 | 0.592 |
| | 336 | 0.622 | 0.534 | 0.417 | 0.428 | 0.417 | 0.428 | 0.869 | 0.677 |
| | 720 | 0.639 | 0.559 | 0.483 | 0.464 | 0.483 | 0.464 | 0.810 | 0.630 |
| | Avg | 0.621 | 0.531 | 0.400 | 0.417 | 0.400 | 0.417 | 0.802 | 0.628 |
| $ETTm_2$ | 96 | 0.264 | 0.352 | 0.201 | 0.286 | 0.201 | 0.286 | 0.352 | 0.454 |
| | 192 | 0.292 | 0.365 | 0.260 | 0.329 | 0.260 | 0.329 | 0.694 | 0.691 |
| | 336 | 0.361 | 0.411 | 0.331 | 0.376 | 0.331 | 0.376 | 2.408 | 1.407 |
| | 720 | 0.515 | 0.490 | 0.428 | 0.430 | 0.428 | 0.430 | 1.913 | 1.166 |
| | Avg | 0.358 | 0.405 | 0.305 | 0.355 | 0.305 | 0.355 | 1.342 | 0.930 |

Table 3: TKG-LLM and other baselines on the ETT dataset: few-shot forecasting results with 5% training data across forecast horizons [96, 192, 336, 720]

| Dataset | Horizon | TKG-LLM | | OFA | | iTransformer | | DLinear | | TimesNet | | PatchTST | | Autoformer | |
|---|---|---|---|---|---|---|---|---|---|---|---|---|---|---|---|
| | | MSE | MAE | MSE | MAE | MSE | MAE | MSE | MAE | MSE | MAE | MSE | MAE | MSE | MAE |
| $ETTh_1$ | 96 | 0.743 | 0.556 | 0.543 | 0.506 | 0.808 | 0.610 | 0.547 | 0.503 | 0.892 | 0.625 | 0.577 | 0.519 | 0.681 | 0.570 |
| | 192 | 0.814 | 0.577 | 0.748 | 0.580 | 0.928 | 0.658 | 0.720 | 0.604 | 0.940 | 0.665 | 0.711 | 0.570 | 0.725 | 0.602 |
| | 336 | 0.957 | 0.665 | 0.754 | 0.595 | 1.475 | 0.861 | 0.984 | 0.727 | 0.945 | 0.653 | 0.816 | 0.619 | 0.761 | 0.624 |
| | 720 | - | - | - | - | - | - | - | - | - | - | - | - | - | - |
| | Avg | 0.838 | 0.599 | 0.681 | 0.560 | 1.070 | 0.710 | 0.750 | 0.611 | 0.925 | 0.647 | 0.694 | 0.569 | 0.722 | 0.598 |
| $ETTh_2$ | 96 | 0.484 | 0.476 | 0.376 | 0.421 | 0.397 | 0.427 | 0.442 | 0.456 | 0.409 | 0.420 | 0.401 | 0.421 | 0.428 | 0.468 |
| | 192 | 0.601 | 0.542 | 0.418 | 0.441 | 0.438 | 0.445 | 0.617 | 0.542 | 0.483 | 0.464 | 0.452 | 0.455 | 0.496 | 0.504 |
| | 336 | 0.710 | 0.598 | 0.408 | 0.439 | 0.631 | 0.553 | 1.424 | 0.849 | 0.499 | 0.479 | 0.464 | 0.469 | 0.486 | 0.496 |
| | 720 | - | - | - | - | - | - | - | - | - | - | - | - | - | - |
| | Avg | 0.598 | 0.539 | 0.400 | 0.433 | 0.488 | 0.475 | 0.694 | 0.577 | 0.439 | 0.448 | 0.827 | 0.615 | 0.441 | 0.457 |
| $ETTm_1$ | 96 | 0.473 | 0.437 | 0.386 | 0.405 | 0.589 | 0.510 | 0.332 | 0.374 | 0.606 | 0.518 | 0.399 | 0.414 | 0.726 | 0.578 |
| | 192 | 0.466 | 0.440 | 0.440 | 0.438 | 0.703 | 0.565 | 0.358 | 0.390 | 0.681 | 0.539 | 0.441 | 0.436 | 0.750 | 0.591 |
| | 336 | 0.500 | 0.464 | 0.485 | 0.459 | 0.898 | 0.641 | 0.402 | 0.416 | 0.786 | 0.597 | 0.499 | 0.467 | 0.851 | 0.659 |
| | 720 | 0.559 | 0.509 | 0.577 | 0.499 | 0.948 | 0.671 | 0.511 | 0.489 | 0.796 | 0.593 | 0.767 | 0.587 | 0.857 | 0.655 |
| | Avg | 0.499 | 0.462 | 0.472 | 0.450 | 0.784 | 0.596 | 0.400 | 0.417 | 0.717 | 0.561 | 0.526 | 0.476 | 0.796 | 0.620 |
| $ETTm_2$ | 96 | 0.230 | 0.309 | 0.199 | 0.280 | 0.265 | 0.339 | 0.236 | 0.326 | 0.220 | 0.299 | 0.206 | 0.288 | 0.232 | 0.322 |
| | 192 | 0.266 | 0.331 | 0.256 | 0.316 | 0.310 | 0.362 | 0.306 | 0.373 | 0.311 | 0.361 | 0.264 | 0.324 | 0.291 | 0.357 |
| | 336 | 0.328 | 0.369 | 0.318 | 0.353 | 0.373 | 0.399 | 0.380 | 0.423 | 0.338 | 0.366 | 0.334 | 0.367 | 0.478 | 0.517 |
| | 720 | 0.472 | 0.454 | 0.460 | 0.436 | 0.478 | 0.454 | 0.674 | 0.583 | 0.509 | 0.465 | 0.454 | 0.432 | 0.553 | 0.538 |
| | Avg | 0.324 | 0.366 | 0.308 | 0.346 | 0.356 | 0.388 | 0.399 | 0.426 | 0.344 | 0.372 | 0.314 | 0.352 | 0.388 | 0.433 |

Table 4: TKG-LLM and other baselines on the ETT dataset: few-shot forecasting results with 10% training data across forecast horizons [96, 192, 336, 720]

| Dataset | Horizon | TKG-LLM | | OFA | | TEST | | iTransformer | | DLinear | | TimesNet | | PatchTST | | Autoformer | |
|---|---|---|---|---|---|---|---|---|---|---|---|---|---|---|---|---|---|
| | | MSE | MAE | MSE | MAE | MSE | MAE | MSE | MAE | MSE | MAE | MSE | MAE | MSE | MAE | MSE | MAE |
| $ETTh_1$ | 96 | 0.749 | 0.554 | 0.458 | 0.456 | 0.455 | 0.457 | 0.790 | 0.586 | 0.492 | 0.495 | 0.861 | 0.628 | 0.516 | 0.485 | 0.613 | 0.552 |
| | 192 | 0.787 | 0.565 | 0.570 | 0.516 | 0.572 | 0.519 | 0.837 | 0.609 | 0.565 | 0.538 | 0.797 | 0.593 | 0.598 | 0.524 | 0.722 | 0.598 |
| | 336 | 0.788 | 0.570 | 0.608 | 0.535 | 0.611 | 0.531 | 0.780 | 0.575 | 0.721 | 0.622 | 0.941 | 0.648 | 0.657 | 0.550 | 0.750 | 0.619 |
| | 720 | 1.010 | 0.692 | 0.725 | 0.591 | 0.723 | 0.594 | 1.234 | 0.811 | 0.986 | 0.743 | 0.877 | 0.641 | 0.762 | 0.610 | 0.721 | 0.616 |
| | Avg | 0.833 | 0.595 | 0.590 | 0.525 | 0.479 | 0.525 | 0.910 | 0.860 | 0.691 | 0.600 | 0.869 | 0.628 | 0.633 | 0.542 | 0.702 | 0.596 |
| $ETTh_2$ | 96 | 0.375 | 0.418 | 0.331 | 0.374 | 0.332 | 0.374 | 0.404 | 0.435 | 0.357 | 0.411 | 0.378 | 0.409 | 0.353 | 0.389 | 0.413 | 0.451 |
| | 192 | 0.419 | 0.448 | 0.402 | 0.411 | 0.401 | 0.433 | 0.470 | 0.474 | 0.569 | 0.519 | 0.490 | 0.467 | 0.403 | 0.414 | 0.474 | 0.477 |
| | 336 | 0.398 | 0.435 | 0.406 | 0.433 | 0.408 | 0.440 | 0.489 | 0.485 | 0.671 | 0.572 | 0.537 | 0.494 | 0.426 | 0.441 | 0.547 | 0.543 |
| | 720 | 0.521 | 0.510 | 0.449 | 0.464 | 0.459 | 0.480 | 0.593 | 0.538 | 0.824 | 0.648 | 0.510 | 0.491 | 0.477 | 0.480 | 0.516 | 0.523 |
| | Avg | 0.428 | 0.453 | 0.397 | 0.421 | 0.401 | 0.432 | 0.489 | 0.483 | 0.605 | 0.538 | 0.479 | 0.465 | 0.415 | 0.431 | 0.488 | 0.499 |
| $ETTm_1$ | 96 | 0.450 | 0.424 | 0.390 | 0.404 | 0.392 | 0.401 | 0.709 | 0.556 | 0.352 | 0.392 | 0.583 | 0.501 | 0.410 | 0.419 | 0.774 | 0.614 |
| | 192 | 0.508 | 0.461 | 0.429 | 0.423 | 0.423 | 0.426 | 0.717 | 0.548 | 0.382 | 0.412 | 0.630 | 0.528 | 0.437 | 0.434 | 0.754 | 0.592 |
| | 336 | 0.525 | 0.423 | 0.469 | 0.439 | 0.471 | 0.444 | 0.735 | 0.575 | 0.419 | 0.434 | 0.725 | 0.568 | 0.476 | 0.454 | 0.869 | 0.677 |
| | 720 | 0.537 | 0.484 | 0.569 | 0.498 | 0.552 | 0.501 | 0.752 | 0.584 | 0.490 | 0.477 | 0.769 | 0.549 | 0.501 | 0.466 | 0.810 | 0.630 |
| | Avg | 0.505 | 0.448 | 0.464 | 0.441 | 0.574 | 0.443 | 0.728 | 0.565 | 0.411 | 0.429 | 0.677 | 0.537 | 0.501 | 0.466 | 0.802 | 0.628 |
| $ETTm_2$ | 96 | 0.209 | 0.291 | 0.188 | 0.269 | 0.233 | 0.262 | 0.245 | 0.322 | 0.213 | 0.303 | 0.212 | 0.285 | 0.191 | 0.274 | 0.352 | 0.454 |
| | 192 | 0.269 | 0.336 | 0.251 | 0.309 | 0.303 | 0.302 | 0.274 | 0.338 | 0.278 | 0.345 | 0.270 | 0.323 | 0.252 | 0.317 | 0.694 | 0.691 |
| | 336 | 0.312 | 0.363 | 0.307 | 0.346 | 0.359 | 0.341 | 0.361 | 0.394 | 0.338 | 0.385 | 0.323 | 0.353 | 0.306 | 0.353 | 2.408 | 1.407 |
| | 720 | 0.412 | 0.417 | 0.426 | 0.417 | 0.452 | 0.419 | 0.467 | 0.442 | 0.436 | 0.440 | 0.474 | 0.449 | 0.433 | 0.427 | 1.913 | 1.166 |
| | Avg | 0.300 | 0.352 | 0.293 | 0.335 | 0.317 | 0.309 | 0.336 | 0.373 | 0.316 | 0.368 | 0.320 | 0.353 | 0.296 | 0.343 | 1.342 | 0.930 |

Table 5: Ablation study on the TKG module

| Ablation Setting | ETTh1-96 | | ETTm1-96 | |
|---|---|---|---|---|
| | MSE | MAE | MSE | MAE |
| w/ TKG | 0.371 | 0.402 | 0.289 | 0.345 |
| w/o TKG | 0.382 | 0.404 | 0.292 | 0.349 |