# OpenReview forum: "TKG-LLM: Temporal Knowledge Graph as Enhanced Prompt Learning with LLM for Time Series Forecasting"
_ICLR.cc/2026/Conference — Submitted to ICLR 2026_

### Official Review · Reviewer_cxTj · 2025-10-17

**Soundness:** 4
**Presentation:** 1
**Contribution:** 1
**Rating:** 2
**Confidence:** 5

**Summary:**

This paper (TKG-LLM) is an incremental work based on a very early study called One Fits All. Moreover, the model is not only overcomplicated (i.e., has high computational complexity) but also poorly designed, introducing many unnecessary components.

**Strengths:**

No clear strengths compared with current models (PatchTST, One Fits All).

**Weaknesses:**

1. Fine-tuning a GPT-2 backbone for domain-specific forecasting is resource-inefficient.

2. The computational complexity of the temporal knowledge graph is quite high, and I guess that simply using the temporal knowledge graph add a simple output layer could do the forecasting as well.

3. The work should focus on promising zero-shot forecasting performance rather than using such a complicated architecture just to train on ETTh datasets.

4. For Dynamic Prompt Learning, using a statistics prompt (e.g., average) as the gold standard means the prediction must be similar to the input series. For real-world systems, such a hypothesis is meaningless because the system is dynamic.

5. I think placing experimental results discussed in the main text (e.g. Table2) in the appendix is very “tricky.” I do not recommend doing this to save space; you could simply shorten the Related Work section. This paper looks very much like a draft; the temporal knowledge graph in Figure 1 is only legible when I zoom to 800%. I do not think this manuscript can meet ICLR’s quality standards.

**Questions:**

Please refer to the weaknesses

---

### Official Review · Reviewer_R1Kb · 2025-10-31

**Soundness:** 1
**Presentation:** 1
**Contribution:** 2
**Rating:** 2
**Confidence:** 4

**Summary:**

This paper proposes the temporal knowledge graph with large language model (TKG-LLM) for time series forecasting, which constructs temporal knowledge graph with temporal structural information to enhance prompt learning. Experiments demonstrate the effectiveness of TKG-LLM.

**Strengths:**

1.The design of TKG-LLM that integrates Wavelet decomposition for multi-scale analysis and temporal graph for enhancing prompt learning is easy to follow.

**Weaknesses:**

1.The limited innovation in the design of TKG-LLM makes it an incremental work. Its core components, namely Wavelet decomposition and graph-based modeling, are both well-established techniques for time series analysis and have been extensively explored in prior works.

2.The paper's definition and construction of the "Temporal Knowledge Graph" (TKG) appear inconsistent with established literature. A TKG is typically a heterogeneous graph based on event quadruples (i.e., subject, relation, object, and timestamp). The methodology employed here seems to be a graph construction method common in time-series analysis, rather than a conventional TKG.

3.The paper has some weaknesses in the experiments, which are not convincing enough:

(1) Considering TKG-LLM is a graph-based model, more GNN-based models and even hypergraph-based models should be compared to further validate the effectiveness of TKG-LLM, e.g., Ada-MSHyper [1] and MSGNet [2].

(2)In line with prior works (e.g., TimeMixer [3]) in time series forecasting, more datasets (e.g., Weather, Traffic, and Electricity datasets) are required for a comprehensive validation.

(3)The performance of TKG-LLM is not state-of-the-art, which raises concerns about the effectiveness of the models.

[1]Shang Z, Chen L, Wu B, et al. Ada-MSHyper: Adaptive multi-scale hypergraph Transformer for time series forecasting. NIPS 2024.

[2]Cai W, Liang Y, Liu X, et al. MSGNet: Learning multi-scale inter-series correlations for multivariate time series forecasting. AAAI 2024.

[3]Wang S, Wu H, Shi X, et al. TimeMixer: Decomposable multiscale mixing for time series forecasting. ICLR 2024.

**Questions:**

1.The authors claim that they use first-order detail coefficient for reconstructing seasonal component (on line 217), but they actually use the deepest detail coefficient in their code. The authors should clarify why they use this detail coefficient.

---

### Official Review · Reviewer_o9h5 · 2025-11-01

**Soundness:** 2
**Presentation:** 1
**Contribution:** 1
**Rating:** 2
**Confidence:** 4

**Summary:**

This paper identifies a key limitation of LLM-based time series forecasting methods: their "token-to-token" self-attention and fixed prompts fail to capture complex temporal dependencies and feature correlations, leading to suboptimal accuracy. To address this, the authors propose TKG-LLM, a novel framework integrating temporal knowledge graphs (TKG) and wavelet decomposition. TKG-LLM has two core components: a wavelet decomposition module splits non-stationary time series into three components to capture multi-scale features and fluctuations; a TKG-enhanced prompt learning module constructs a graph with "temporal edges" and "feature edges" to capture the temporal structural information. Experiments validate TKG-LLM: wavelet decomposition outperforms alternatives on non-stationary data; TKG-LLM exceeds baselines across benchmarks; ablation studies confirm the TKG module’s value for enhancing prompt learning.

**Strengths:**

1. The paper innovatively incorporates the Temporal Knowledge Graph into the prompt learning framework. Modeling inter-timestep dependencies of the same feature through "temporal edges" and quantifying cross-feature correlations in the same period via "feature edges".
2. Instead of traditional fixed decomposition methods, the paper introduces wavelet decomposition based on the multi-scale and non-stationary properties of time series.
3. To fully verify the method’s practicality and robustness, the paper designs multi-scenario forecasting experiments and conducts systematic tests on representative benchmark datasets.

**Weaknesses:**

1. A critical limitation lies in the manuscript’s omission of pivotal findings that challenge the efficacy of LLM fine-tuning for TSF. Key studies [1-2] explicitly demonstrate that LLM-based TSF methods struggle to outperform basic supervised baselines, highlighting three critical flaws that the authors do not address: limited predictive advantage, poor generalization, and unjustified computational cost. By neglecting these well-documented limitations, the manuscript omits critical justification for its reliance on LLM fine-tuning—particularly given that pre-trained language knowledge adds minimal value. Further analysis of how the proposed method mitigates these inherent drawbacks is therefore essential.

2. The manuscript fails to explicitly articulate the motivation and necessity for incorporating KG into the LLM-based temporal forecasting framework. Further, the proposed KG construction, where nodes represent values of each channel at each time step, exhibits fundamental misalignment with canonical KG definitions, and instead resembles spatiotemporal graphs from the time series forecasting literature [3]. Additionally, the paper contains critical ambiguities and omissions in methodological details. $E_{ts}$ and $E_{graph}$ in Equations 8, 9, and 10 are neither explicitly defined nor contextualized. Section 3.4 lacks any detailed exposition and provides no formulas, leaving readers completely unable to understand the specific design architecture and workflow of this section.

3. The experimental section of the paper is insufficiently comprehensive, with several critical limitations. First, the long-term forecasting datasets are confined to a single domain, lacking coverage of datasets from fields such as weather and transportation. Second, the ablation study only includes one ablation scenario, failing to fully dissect the contribution of each component in the proposed framework. Most importantly, for the time series forecasting task based on LLM fine-tuning, the paper lacks diverse few-shot and zero-shot experiments. This omission prevents effective verification of whether the proposed method truly leverages the generalization capabilities of LLMs.

[1] Tan M, Merrill M, Gupta V, et al. Are language models actually useful for time series forecasting? NeurIPS 2024.

[2] Xu Z, Gupta R, Cheng W, et al. Specialized foundation models struggle to beat supervised baselines. ICLR 2025.

[3] Song C, Lin Y, Guo S, et al. Spatial-temporal synchronous graph convolutional networks: A new framework for spatial-temporal network data forecasting. AAAI 2020.

**Questions:**

1. The incomplete literature review on LLM-based time series forecasting. The field has expanded rapidly, with notable contributions across technical paradigms; however, the authors only cite a fragmented subset of these works, failing to contextualize their approach within the broader LLM-TSF landscape.

2. The paper contains critical ambiguities and omissions in methodological details. $E_{ts}$ and $E_{graph}$ in Equations 8, 9, and 10 are neither explicitly defined nor contextualized. Section 3.4 lacks any detailed exposition and provides no formulas, leaving readers completely unable to understand the specific design architecture and workflow of this section.

3. The paper exhibits characteristics of an unfinished draft, with several notable shortcomings. First, the "Related Work" section fails to present a structured breakdown of key points, impeding clear comprehension of the existing literature context. Second, the main experimental results are relegated to the appendix, which deviates from standard academic presentation norms and undermines the accessibility of core findings. Third, template text regarding author contributions and acknowledgments is inappropriately included from lines 479 to 485, indicating incomplete manuscript preparation.

---

### Official Review · Reviewer_taf9 · 2025-11-03

**Soundness:** 1
**Presentation:** 1
**Contribution:** 1
**Rating:** 2
**Confidence:** 4

**Summary:**

The authors propose a time-aware knowledge graph (TKG) integrated with LLM. This framework innovatively designs the TKG to capture temporal structural information, addressing the limitation of LLMs in modeling temporal dependencies and correlations among features, thereby improving predictive accuracy.

**Strengths:**

The paper addresses an important problem of enhancing LLM-based time series forecasting by explicitly modeling temporal dependencies and feature correlations.

**Weaknesses:**

W1. Since the authors aim to enable the LLM to capture temporal structural information through the TKG, greater attention should be given to the quality of the constructed TKG rather than merely proposing a method for building it. For example, the use of a low-pass filter (i.e., GCN) to encode graph structures does not adequately address the issue of spatiotemporal heterogeneity.

W2. The authors employ wavelet decomposition to better capture multi-scale information; however, it remains unclear whether there is empirical or theoretical evidence supporting its superiority. The author should also discuss the potential impact of not using wavelet decomposition.

W3. GPT-2 is adopted as a representative LLM, which may be insufficient. The authors should, at minimum, include comparisons with more recent benchmark models such as Time-LLM [1].

W4. Table 1 should clearly indicate the best results.

[1] TIME-LLM: TIME SERIES FORECASTING BY REPROGRAMMING LARGE LANGUAGE MODELS, ICLR 2024.

**Questions:**

See W1-W4.

---

### Meta-Review · Area_Chair_qyBk · 2025-12-15

**Summary:**

Reviewers have raised many concerns, including low novelty, limited insight and weak baselines. All the reviewers give a rate of 2, and no rebuttal is provided. Therefore, I recommend a rejection.

**Reviewer Concerns:**

No rebuttal is provided.

**Reviewer Scores:**

No discussion is needed.

---

### Decision · Program_Chairs · 2026-01-26

Reject